# Application of Ozone Treatment for the Decolorization of the Reactive-Dyed Fabrics in a Pilot-Scale Process—Optimization through Response Surface Methodology

**Ajinkya Sudhir Powar** [1,2,3,4,*] 🆔, **Anne Perwuelz** [1], **Nemeshwaree Behary** [1], **Levinh Hoang** [5] **and Thierry Aussenac** [5]

1.  Ecole Nationale Supérieure des Arts et Industries Textiles (ENSAIT), GEMTEX Laboratory, 2 allée Louise et Victor Champier BP 30329, 59056 Roubaix, France; anne.perwuelz@ensait.fr (A.P.); nmassika.behary@ensait.fr (N.B.)
2.  University de Lille, Nord de France, F-59000 Lille, France
3.  Faculty of Textiles Leather and Industrial Management, Gheorghe Asachi Technical University of Iasi, B-dul. D. Mangeron Nr. 67, 700050 Iasi, Romania
4.  College of Textile and Clothing Engineering, Soochow University, Suzhou 215000, China
5.  Institut Polytechnique UniLaSalle, Transformations & Agro-Ressources EA7519, BP30313, 60026 Beauvais, France; Levinh.Hoang@unilasalle.fr (L.H.); thierry.aussenac@unilasalle.fr (T.A.)
*   Correspondence: ajinkya.powar@ensait.fr; Tel.: +33-758-635-791

**Abstract:** The decolorization of a cotton fabric dyed with a reactive dye (C.I. Reactive Black 5) was studied using an optimized ozone-assisted process at pilot scale. Box–Behnken design was used to evaluate the effects of three parameters on the decolorization of the dyed textile, namely, pH of the treatment (3–7), ozone concentration (5–85 g/m$^3$ of ozone), and treatment time (10–50 min). The fitted mathematical model allowed us to plot response surfaces as well as isoresponse curves and to determine optimal decolorization conditions. In this study, we have proposed a pilot-scale machine which utilizes ozone for the color stripping of the dyed cotton. This pilot-scale application opens up the route for application of ozone at an industrial scale for achieving sustainability in the textile industry.

**Keywords:** box–behnken design; reactive dyed cotton fabric; ozone-assisted process; color removal; mechanical properties; pilot-scale approach; response surface methodology

## 1. Introduction

In the textile wet processing industry, the primitive approach for correcting the off-shade problems in dyed textiles is chemical stripping by the use of oxidizing or reducing agents [1]. Alkaline reductive stripping is a commonly preferred method for color stripping in the case of cotton fabrics dyed with reactive dyes. The chemical reaction occurring consists in the cleavage of chemical bonds of the chromophore group, responsible for the color [2]. Sodium hydrosulphite is the most commonly used reducing agent in alkaline reductive application used in the conventional stripping method. Still, there are various drawbacks associated with the use of this chemical, since it is comparatively unstable and decomposes readily in aqueous form and in the presence of oxygen. In wet textile processing, normally huge quantities of sodium hydrosulphite are employed for color stripping application. Sodium hydrosulphite is toxic, and upon contact with moisture, it is oxidized to hydrogen sulfite, sulfite, and hydrogen sulfate, and under strongly acidic conditions, it may liberate sulfur dioxide,

which is known to induce respiratory irritation in humans [3]. Moreover, the stripping chemicals cause damage to the textile [2,4]. Hence an ecofriendly and sustainable method for the dye stripping of reactive-dyed cotton needs to be established as a substitute for the conventional stripping method.

Ozone is a strong oxidizing agent having an oxidation potential (of 2.07 V) higher than other oxidizing agents available today in the market. Application of ozone in several domains of textile finishing has been reported. These include reduction clearing of polyester dyed with disperse dyes [5], discharge printing of reactive-dyed cotton fabrics [6], color stripping of reactive dyes from cotton textiles [7], and bleaching of raw cotton textiles [8]. The industrial application of ozone has been limited to denim and garment washing only, and its application to other areas can be a scope of study [9].

Utilization of ozone for pilot scale color stripping of the reactive-dyed textiles has, however, never been reported and this work studies the impact of several process parameters (pH, ozone concentration, and treatment time) on pilot scale color stripping of cotton fabrics dyed with a reactive dye (C.I. Reactive Black 5).

Ozone is generally produced by electrical discharge of gaseous oxygen. For ozone application in the liquid medium (i.e., water treatment or wastewater treatment), ozone transfer from gas phase to liquid phase is a necessary process to yield dissolved ozone in water. Different methods of gas dispersion are applied in practice and the most popular are diffusers, static mixer, and injection [10].

In this study, we have proposed a pilot scale machine which utilizes ozone for the color stripping of the dyed cotton. This pilot scale application opens up the route for the application of ozone at the industrial scale for achieving sustainability in the industry. The contact between the two phases accompanied by an ozone mass transfer in this pilot is based on the venturi injection method.

Experimental design methodology is a good strategy that makes simultaneous variation of all experimental variables feasible and gives information to optimize processes. Thanks to statistical analysis of the generated data, remarkable information is provided on the interactions among the experimental variables. From here, the number of tests and the required time would be reduced, leading to a considerable reduction in the overall required cost. Thus, Box–Behnken design was used. It is a cubic, independent quadratic and rotatable design with the treatment combinations at the midpoints of the edges of a multidimensional cube without embedded factorial or fractional factorial design and is used for fitting second-order response surfaces.

## 2. Materials and Methods

### 2.1. Materials

A 100% ready for dyeing (RFD) woven cotton fabric (GSM 150) procured from France was used for the experimentation. Noir Everzol B, C.I. Reactive Black 5 dye supplied by the Achitex Minerva, France, a commonly used azo dye in the textile industry, was used for the dyeing experiments (Figure 1).

**Figure 1.** Structure of C.I. Reactive Black 5 dye.

### 2.2. Dyeing Procedure for the Cotton Fabrics

The dyeing process was carried out in a Jigger machine (Teinturerie Lenfant, France). Dyestuff of 1% of fabric weight was used with the standard recipe for the dyeing of cotton fabric with reactive dye (C.I. Reactive Black 5).

### 2.3. Color Removal from the Dyed Cotton Textiles

The dyed material was divided and cut into small rectangular samples with dimensions of 30 cm × 18 cm for the color stripping experiments. A total of 40 g of dyed textile was treated for each stripping experiment.

Three different pH values (3, 5, 7) were used for color removal. The pH of solutions was adjusted using different concentrations of phosphoric acid and/or caustic soda (laboratory grade). The ozone flow rate was set up to three different values (5, 45, 85 $g/m^3$ NTP ozone) and three treatment times (10, 30, 50 min) were chosen. All the ozonation treatments were performed with the help of reverse osmosis water and carried out at room temperature in an ozonation reactor with liquor amounting to 60 liters.

### 2.4. Ozonation Treatment

The ozonation process carried out on an ozone platform (UniLasalle, France) is shown in Figure 2. The system included two parts: (i) the ozone generator and (ii) the ozonation reactor. Ozone was generated from pure oxygen by Corona discharge (Ozat CFS3-2G, Ozonia, France) with measurement of the ozone concentration in the inlet and in off-gas (BMT, Germany). In comparison with ozone produced using dry air, ozone generated from oxygen can produce higher concentrations of ozone with the same ozonator.

Ozone gas was injected in the reactor using a venturi system combined with a circulation pump. The system allowed ozone gas to be transferred in dissolved form for the decolorization process. In comparison to ozone diffused using a dome ceramic bubble column, the venturi system is more advantageous for the smaller reactor used in this study.

The dyed textile fabric to be decolorized was placed vertically in the ozonation reactor consisting of the osmosis water. After the completion of the ozonation treatment, the treated textile was washed with normal tap water in order to eliminate any residual ozone and some of the byproducts of oxidation. Finally, the textile fabric was dried for colorimetric analysis and mechanical testing.

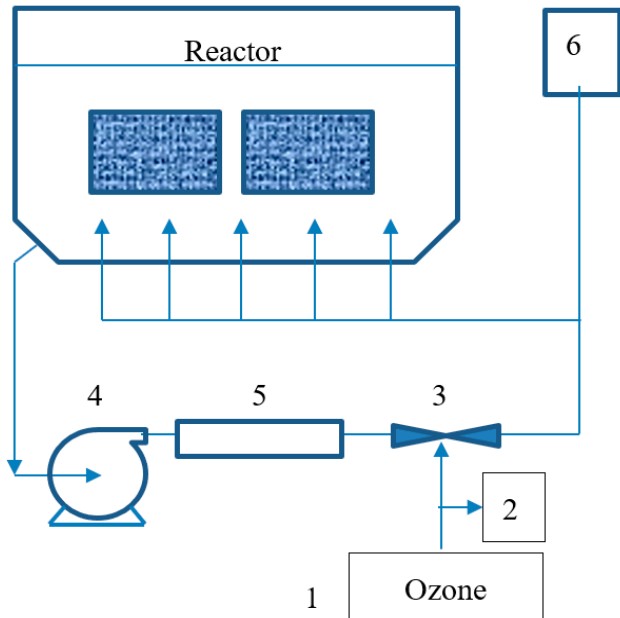

**Figure 2.** *Cont.*

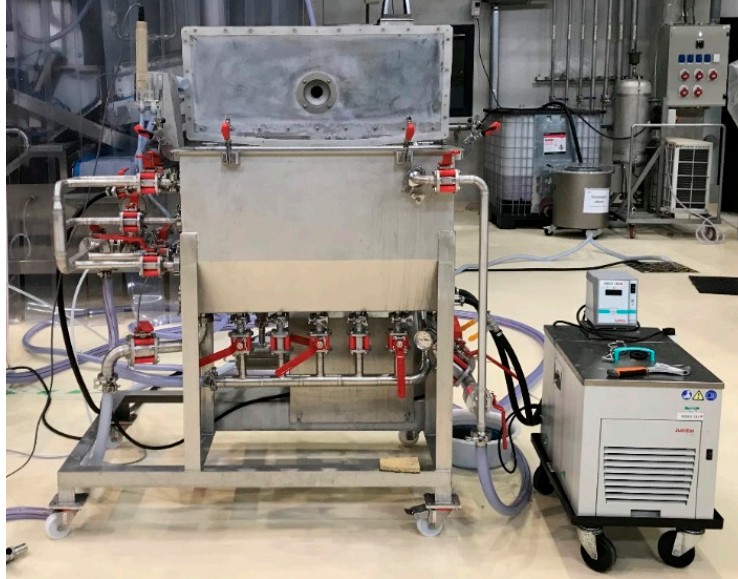

**Figure 2.** Pilot ozonation (1: ozone generator; 2: ozone analyzer; 3: venturi injection system; 4: circulation pump; 5: filter; 6: dissolved ozone analyzer and pH meter).

*2.5. Color Stripping Percentage Measurements*

After the color stripping processes, the visual color of each substrate was measured using a Konica Minolta cm3600d spectrophotometer (Konica Minolta Inc., Tokyo, Japan). The fabric reflectance was measured for wavelength varying from 350 nm to 750 nm, and the color intensity '$K/S$ value' was calculated from the reflectance values using the Kubelka–Munk equation [11]. Figure 3 shows the $K/S$ spectral curve of the cotton dyed fabric without stripping, with a main peak at 610 nm, and a small peak at 400 nm.

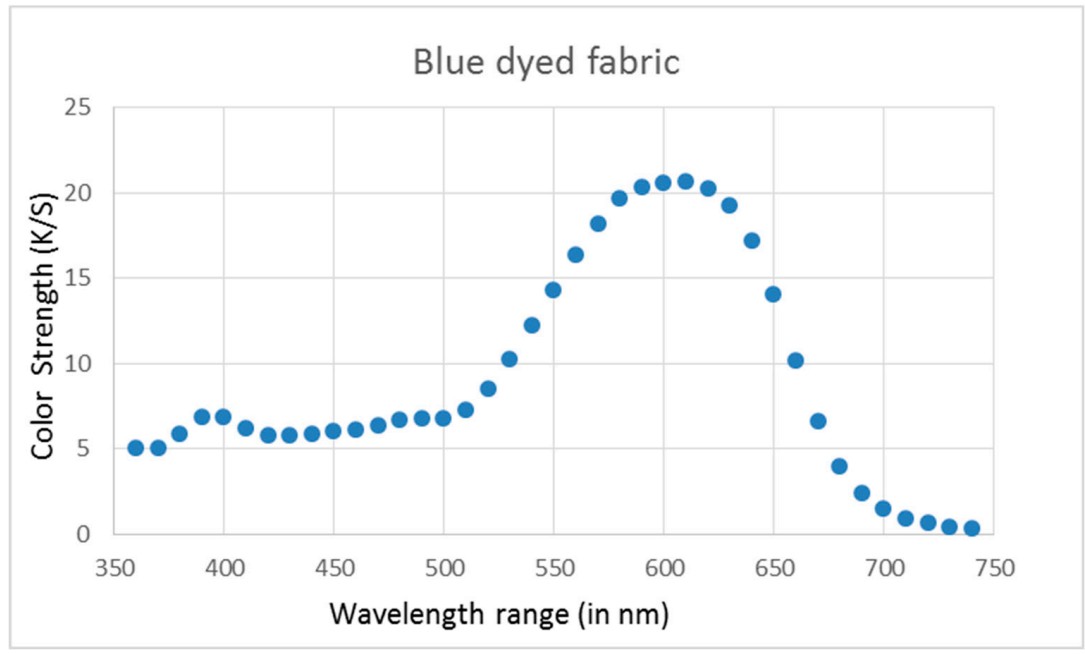

**Figure 3.** $K/S$ spectrum of the blue dyed fabric.

On the decolorized samples, the $K/S$ values were determined from 20 different scanned areas of each fabric sample. The measurements were taken on both sides of the fabric for consistency, and then

the average *K/S* value was calculated. The color stripping percentage was calculated at maximum intensity wavelength (λmax = 610 nm) using the following formula [12]:

$$Stripping\ percent = \frac{\left[\frac{K}{S}\ value\ of\ dyed\ sample - \frac{K}{S}\ value\ of\ stripped\ sample\right]}{\frac{K}{S}\ value\ of\ dyed\ sample} \times 100$$

### 2.6. Mechanical Strength Tests

The mechanical properties of the color-stripped fabrics were carried out using the MTS (2/M) automatic testing system. The cotton fabric samples were cut into rectangular dimensions of 30 cm × 5 cm, and the tensile properties were measured only in one direction (weftwise) following the international standards for tensile properties NF EN ISO 13934-1; 2013.20 Tests were carried out on five samples for unstripped and stripped samples, and the average tensile strength was calculated.

### 2.7. Method: Box–Behnken Tool for the Design of the Experiments

The Box–Behnken design was used as experimental design tool for the response surface methodology to optimize color stripping by ozone. The influence of pH on the color stripping was investigated in the range of 3–7 using diluted solutions of $H_3PO_4$/NaOH. The concentration of ozone was varied from 5–85 $g/m^3$ using varying power of the ozone generator and the influence of reaction time in the range of 10 to 50 min was also studied.

The number of experiments (*N*) based on the color stripping by ozone is defined by the following expression:

$$N = 2K\,(K-1) + C_0 = 2*3*(3-1) + 4 = 16$$

where *K* denotes the number of variables and $C_0$ denotes the number of replications at the center point. In this study, *K* and $C_0$ were set to 3 and 4, respectively. Hence, 16 experiments had to be done to perform a Box–Behnken design. The levels of each variable are listed in Tables 1 and 2.

**Table 1.** Box–Behnken design experimental plan 3 factors * 3 levels.

| Experiment No. | pH | Ozone Concentration (g/m³ TPN) | Time (Min) |
|:---:|:---:|:---:|:---:|
| F | X1 | X2 | X3 |
| E1 | 5 | 5 | 10 |
| E2 | 7 | 45 | 10 |
| E3 | 5 | 85 | 10 |
| E4 | 3 | 45 | 10 |
| E5 | 3 | 5 | 30 |
| E6 | 7 | 5 | 30 |
| E7 | 7 | 85 | 30 |
| E8 | 3 | 85 | 30 |
| E9 | 5 | 5 | 50 |
| E10 | 7 | 45 | 50 |
| E11 | 5 | 85 | 50 |
| E12 | 3 | 45 | 50 |
| E13 | 5 | 45 | 30 |
| E14 | 5 | 45 | 30 |
| E15 | 5 | 45 | 30 |
| E16 | 5 | 45 | 30 |

The responses can be modeled by a second-order polynomial equation:

$$Y = a_0 + a_1 x_1 + a_2 x_2 + a_3 x_3 + a_{12} x_1 x_2 + a_{13} x_1 x_3 + a_{23} x_2 x_3 + a_{11} x_1^2 + a_{22} x_2^2 + a_{33} x_3^2$$

where $Y$ is the color-stripping percentage varying as a function of $x_1$ (pH), $x_2$ (concentration of ozone), and $x_3$ (reaction time) variables. $a_0$ is the intercept, $a_1$, $a_2$, $a_3$ to $a_{11}$; $a_{22}$ and $a_{33}$ are the regression coefficients.

**Table 2.** Experimental conditions.

| Factor Level | Lower | Central | Upper |
|---|---|---|---|
| | **−1** | **0** | **+1** |
| pH | 3 | 5 | 7 |
| Concentration ozone g/m$^3$ TPN | 5 | 45 | 85 |
| time (minutes) | 10 | 30 | 50 |

In this study, Excel (Microsoft Office) was used to analyze data and calculate the predicted responses of the experimental design.

## 3. Results

### 3.1. Experimental Results for Different Decolorization Experiments

#### 3.1.1. *K/S* Spectral Analysis of Samples Stripped Using Experiment E13

A dyed cotton sample stripped with the E13 experimental conditions (pH 5, 45 g/m$^3$ of ozone, and treatment time of 50 min) was analyzed with a spectrophotometer. The fabric sample was scanned 20 times in 20 different areas of the same sample.

Figure 4 shows the different spectral curves before and after stripping. A strong decrease of the *K/S* at all wavelengths can be observed. The deviation between all the experiments is almost similar for *K/S* values measured at each wavelength, from $\lambda = 350$ to $\lambda = 650$ nm. The variation coefficient of the *K/S* values is highest at $\lambda = 610$ nm and is equal to 21%. That means that the dyed sampled has been stripped but there is still a small deviation due to heterogeneity of the ozone treatment on the sample.

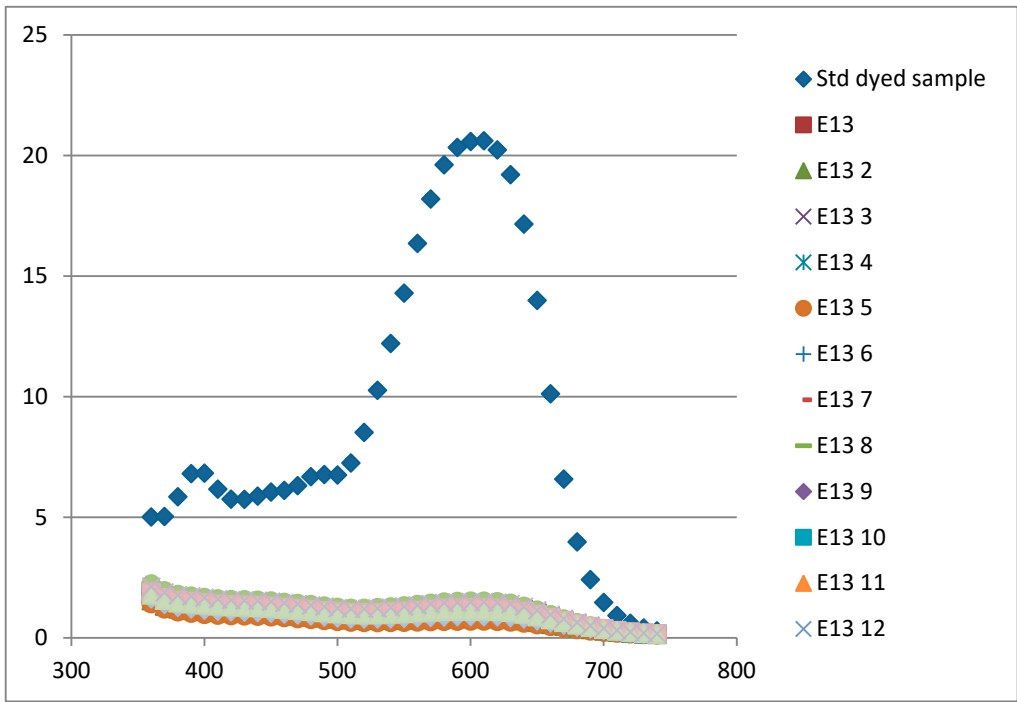

**Figure 4.** *K/S* spectral curves of stripped fabric scanned 20 times in 20 different areas, compared to that of the dyed cotton fabric.

### 3.1.2. Reproducibility of the Experiments

Four experiments, E13 to E16, were carried out using the same conditions as E13 (pH = 5, ozone flow = 45 g/m$^3$, time = 30 min) on four different samples of dyed fabric.

From the graph (Figure 5), we can assess the repeatability of the central point of the experiments of the Box–Behnken design. All the experiments (E13, E14, E15, and E16) show the same values of *K/S* values variation. *K/S* values of stripped samples are close to that of the undyed sample. This means that these stripped samples can be used as potential alternatives for virgin (undyed) cotton, in the framework of textile processes and products.

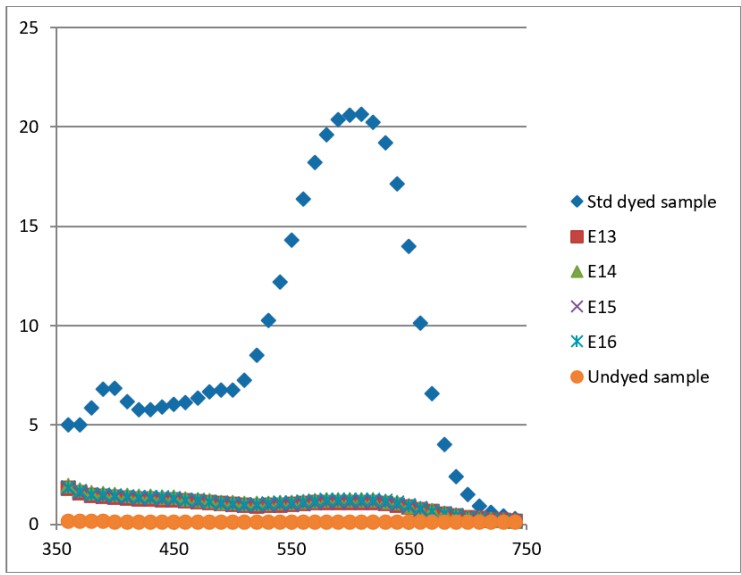

**Figure 5.** Reproducibility of the central point of the experiments.

### 3.1.3. Colorimetric Analysis of Fabrics Treated with 16 Experimental Conditions (E1 to E16)

*K/S* spectral curves of fabrics stripped using 16 different experimental conditions are shown in Figure 6. Considerable variation of *K/S* values, and hence of stripping treatment, is observed as a function of experimental conditions used.

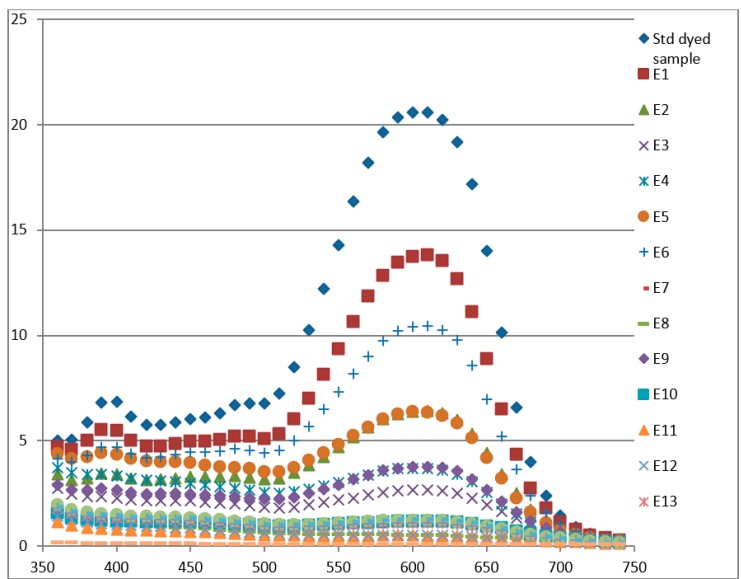

**Figure 6.** *K/S* vs wavelength for all the decolorization experiments.

For experiments E1, E2, and E6, the absorbance peak of each spectral curve at 610 nm is still high (*K/S* > 5), and the stripped samples are still blue (Figure 7B,C,G). Small quantities of ozone combined with low treatment time or higher pH (7) are inefficient in yielding a good stripping.

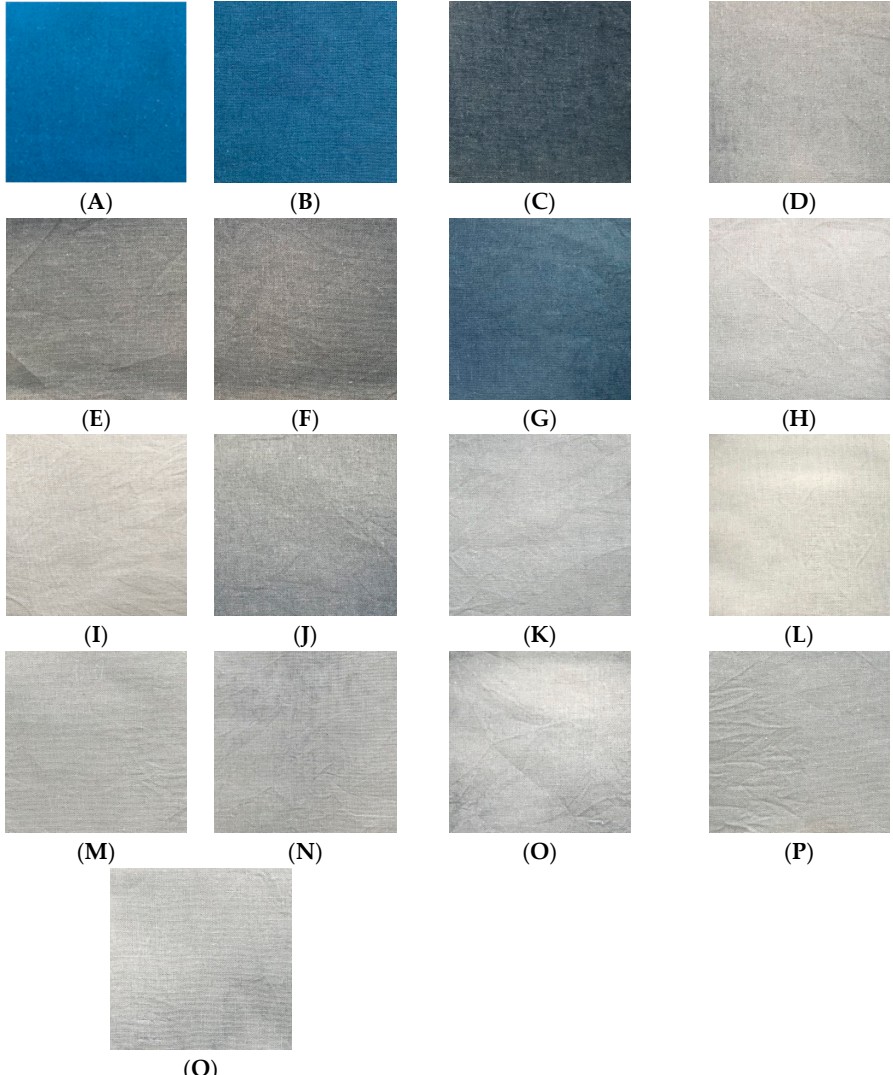

**Figure 7.** Images of the standard dyed fabric and the ozone-treated samples with different experimental parameters: (**A**) standard dyed fabric, (**B**) E1, (**C**) E2, (**D**) E3, (**E**) E4, (**F**) E5, (**G**) E6, (**H**) E7, (**I**) E8, (**J**) E9, (**K**) E10, (**L**) E11, (**M**) E12, (**N**) E13, (**O**) E14, (**P**) E15, (**Q**) E16.

The *K/S* values at 400 nm of the stripped samples are between 1 and 5, which are higher than that of the undyed fabric sample (*K/S* = 0.2) but much smaller than that of the dyed blue reference sample (*K/S* = 7).

For the best stripping treatment, *K/S* is less than 1.5 for all wavelengths and the stripped fabric is almost white.

For certain stripping experiments, the *K/S* value at 360 nm (near the UV region) increases after stripping. This can be related to the absorbance by aromatic groups in the degraded dyestuff molecules, which stay in the fabric after the stripping experiment.

### 3.2. Color Analysis of the Decolorization Samples

The fabric decolorizing was observed directly through photography of the stripped samples (Figure 7).

The CIELAB methodology was applied to further analyze the color of the stripped fabrics (Table 3). The CIELAB color space (also known as CIE L*a*b* or sometimes abbreviated as simply "Lab" color space) is a color space defined by the International Commission on Illumination (CIE) in 1976. It expresses color as three values: L* for the lightness from black (0) to white (100), a* from green (−) to red (+), and b* from blue (−) to yellow (+). CIELAB was designed so that the same amount of numerical change in these values corresponds to roughly the same amount of visually perceived change.

**Table 3.** Colorimetric values (L*, a*, b*, dL*, da*, db*, dE*) for the dyed, undyed, and decolorization experiments E1, E5, E8, E9, E11.

| Description | L* (D65) | a* (D65) | b* (D65) | dL* (D65) | da* (D65) | db* (D65) | dE*ab (D65) | dE cmC (l:c) (D65) |
|---|---|---|---|---|---|---|---|---|
| Std dyed sample | 23.41 | −3.81 | −14.7 | —— | —— | —— | —— | —— |
| Std undyed RFD fabric | 83.76 | −0.08 | 0.47 | —— | —— | —— | —— | —— |
| E1 | 28 | −5.1 | −12.01 | 4.59 | −1.29 | 2.69 | 5.49 | 7.28 |
| E5 | 36.07 | −4.91 | −4.27 | 12.66 | −1.1 | 10.42 | 16.45 | 20.59 |
| E8 | 66.61 | −0.46 | 8.77 | 43.2 | 3.35 | 23.46 | 49.28 | 68.2 |
| E9 | 43.3 | −5.08 | −3.84 | 19.89 | −1.27 | 10.86 | 22.71 | 30.73 |
| E11 | 67.98 | −1.1 | 5.91 | 44.57 | 2.71 | 20.61 | 49.18 | 68.84 |

We can clearly observe that, among all stripping experiments, only E8 and E11 led to higher L* values (L*~68) compared to the unstripped dyed fabric (L* = 23), but these values are lower than that of an undyed white sample (L = 84) (Table 3). Also, the a* values are closer to that of the undyed fabric in these two experiments. However, for these experiments, there was increased yellowness in the fabric (b* = 9 and 5, for E8 and E11, respectively) which can be related to the peak observed at 360 nm (Figure 6).

## 3.3. Tensile Strength Results

For each of the 16 experiments (E1 to E16), tensile strength of 10 different stripped samples was measured and the average value and standard deviation determined (Table 4). It can be seen that for all stripping conditions, there was a decrease in the fabric tensile strength, which is a usual phenomenon observed in chemical color-stripping processes. For experiments E8 and E11, tensile strength loss was almost 28% and 36%, respectively.

**Table 4.** Tensile strength and CV % of the standard and the ozone-treated samples.

| Sr.No. | Tensile Strength (*N*) | CV % (Coefficient of Variation) |
|---|---|---|
| Std sample | 386 | 5 |
| E1 | 295 | 15 |
| E2 | 298 | 16 |
| E3 | 326 | 10 |
| E4 | 330 | 11 |
| E5 | 320 | 10 |
| E6 | 325 | 4 |
| E7 | 292 | 9 |
| E8 | 278 | 13 |
| E9 | 281 | 8 |
| E10 | 284 | 15 |
| E11 | 244 | 13 |
| E12 | 292 | 6 |
| E13 | 288 | 11 |
| E14 | 273 | 9 |
| E15 | 288 | 10 |
| E16 | 329 | 4 |

### 3.4. Box–Behnken Results and Optimization

The values of color stripping (%) and tensile strength loss (%) for each experiment are given in Table 5.

**Table 5.** Experimental design and observed response.

| Sr.No. | pH | Concentration Ozone (g/m$^3$ TPN) | Time (min) | Color Stripping % | Tensile Strength Loss % |
|---|---|---|---|---|---|
| Std sample | X1 | X2 | X3 | N.A. | N.A. |
| E1 | 5 | 5 | 10 | 33.00 | 24 |
| E2 | 7 | 45 | 10 | 68.98 | 23 |
| E3 | 5 | 85 | 10 | 87.10 | 16 |
| E4 | 3 | 45 | 10 | 82.12 | 15 |
| E5 | 3 | 5 | 30 | 69.19 | 17 |
| E6 | 7 | 5 | 30 | 49.19 | 16 |
| E7 | 7 | 85 | 30 | 95.11 | 25 |
| E8 | 3 | 85 | 30 | 97.45 | 28 |
| E9 | 5 | 5 | 50 | 81.62 | 27 |
| E10 | 7 | 45 | 50 | 94.30 | 27 |
| E11 | 5 | 85 | 50 | 97.56 | 37 |
| E12 | 3 | 45 | 50 | 97.48 | 25 |
| E13 | 5 | 45 | 30 | 94.62 | 26 |
| E14 | 5 | 45 | 30 | 94.10 | 29 |
| E15 | 5 | 45 | 30 | 94.65 | 25 |
| E16 | 5 | 45 | 30 | 93.93 | 15 |

### 3.4.1. Adequacy of the Model

The effect of experimental variables on color stripping was investigated thoroughly by considering individual variables (the pH, ozone concentration, and reaction times) along with nonlinear and interaction effects (Table 6).

**Table 6.** Regression statistics.

| | |
|---|---|
| Coefficient of multiple determination | 0.9955 |
| Coefficient of determination R^2 | 0.9910 |
| Coefficient of determination R^2 | 0.9730 |
| Standard Error | 3.1379 |
| Observations | 16 |

For each run, the color-stripping percentage was used as the experimental response. By using analysis of variance (ANOVA/Excel/Microsoft Office 2013), the significance of each term in the quadratic model was assessed (Table 7).

**Table 7.** Analysis of variance (ANOVA) for the refined model.

| Source | Degree of Liberty | Sum of Squares | Average Squares | F | Critical Value of F |
|---|---|---|---|---|---|
| Regression | 10 | 5419 | 541.9 | 55.0 | 0.00017586 |
| Residues | 5 | 49 | 9.9 | | |
| Total | 15 | 5468 | | | |

The lack-of-fit test is designed to define whether the selected model is adequate to describe the observed data or a more complicated model is required. The test is performed by comparing the variability of the current model residuals to the variability between observations at replicate settings of the variables. Based on the ANOVA table, the *p*-value for lack-of-fit was found to be less than 0.05, thus the model appears to be adequate for the observed data at the 95.0% confidence level (Table 8).

**Table 8.** Parameter estimation: lack-of-fit test.

| Term | Estimated Value | Standard Error | Statistical t | Probability (p) |
|---|---|---|---|---|
| Constant | 94.325335 | 1.5689357 | 60.12059 | 0.0000001 |
| $X_1$ | −4.832944 | 1.1094051 | −4.35634 | 0.0073154 |
| $X_2$ | 18.028194 | 1.1094051 | 16.25033 | 0.0000161 |
| $X_3$ | 12.469954 | 1.1094051 | 11.24022 | 0.0000973 |
| $X_1X_2$ | 4.4139817 | 1.5689357 | 2.81336 | 0.0373999 |
| $X_1X_3$ | 2.4886159 | 1.5689357 | 1.586181 | 0.1735572 |
| $X_2X_3$ | −9.5422213 | 1.5689357 | −6.08197 | 0.0017372 |
| $X_1X_1$ | −2.8435008 | 1.5689357 | −1.81238 | |
| $X_2X_2$ | −13.745048 | 1.5689357 | −8.76075 | 0.0003212 |
| $X_3X_3$ | −5.7590606 | 1.5689357 | −3.67068 | 0.0144331 |

### 3.4.2. ANOVA and Multiple Nonlinear Regression Results

The R-squared statistic demonstrated that the fitted model explains 99.55% of the variability in color-stripping percentage (Table 6).

In the current study, the adjusted $R^2$ obtained is 0.9910, which is within the acceptable limit of $R^2$ ≥ 0.80, demonstrating a good fitting of the experimental data with second-order polynomial equation.

$$Y = 94.33 - 4.83x_1 + 18.03x_2 + 12.47x_3 + 4.41x_1x_2 + 2.49x_1x_3 - 9.54x_2x_3$$
$$-2.84x_1^2 - 13.75x_2^2 - 5.76x_3^2$$

Coefficients with more than one factor term and those with second-order terms represent interaction terms and quadratic relationships, respectively. The sign of each coefficient shows how the related factor influences the response. If the coefficient is positive, the response increases (synergetic effect), and if it is negative, the response decreases (antagonist effect).

The relationship between the dependent and independent variables was further elucidated using response surface (3D) methodology.

### 3.5. Optimization of Color Stripping

The quadratic model obtained from the Box–Behnken design generates response surface images for mutual interactive effects as a function of two variables while another variable is kept constant (Figures 8 and 9). According to the coefficient factors, the concentration of ozone and reaction time have dominant positive effects in comparison with the effect of pH on the color stripping. It means that higher ozone concentration and longer reaction time lead to a better efficiency of color stripping. On the contrary, color stripping decreases with increasing pH.

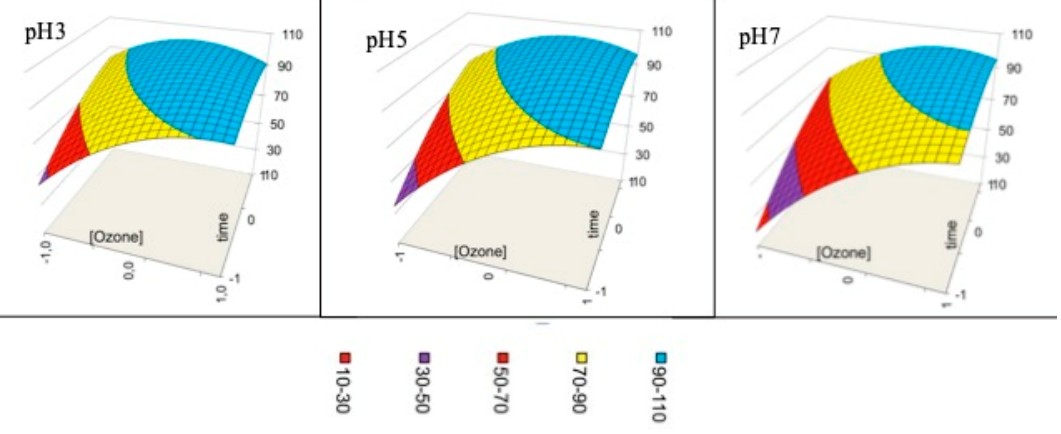

**Figure 8.** Response surface, effect of ozone concentration and reaction time on color stripping of dyed fabrics at different pH (color of the graphs are related to color stripping %).

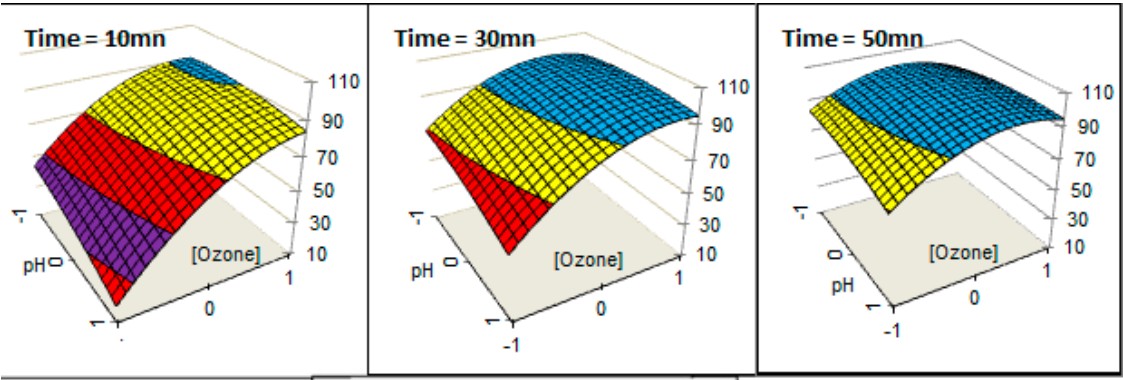

**Figure 9.** Response surface, effect of pH and ozone concentration on the color stripping at different reaction times: 10 min; 30 min; and 50 min.

The optimum predicted point for the maximum decolorization was obtained at pH 5, using an ozone concentration of 85 g/m$^3$ and treatment time of 50 min.

### 3.6. Optimization of the Mechanical Properties

The same optimization methodology was used for the mechanical strength losses. Statistical results were less accurate than those obtained for color stripping. However, the R-squared value was 90.03% and the adjusted R$^2$ = 0.8105, showing the good fitting of the polynomial equation:

$$Y = 26.79 + 0.70x_1 + 2.67x_2 + 4.82x_3 - 0.55x_1x_2 - 1.59x_1x_3 + 4.33x_2x_3 - 4.60x_1^2 \\ -0.84x_2^2 - 0.07x_3^2$$

Figure 10 shows that the tensile strength loss increased with an increase in all variables (pH, ozone concentration, and time). The optimum condition was obtained when the tensile strength loss was at a minimum, that is, when pH = 3, with a concentration of ozone of 5 g/m$^3$ and treatment time of 10 min.

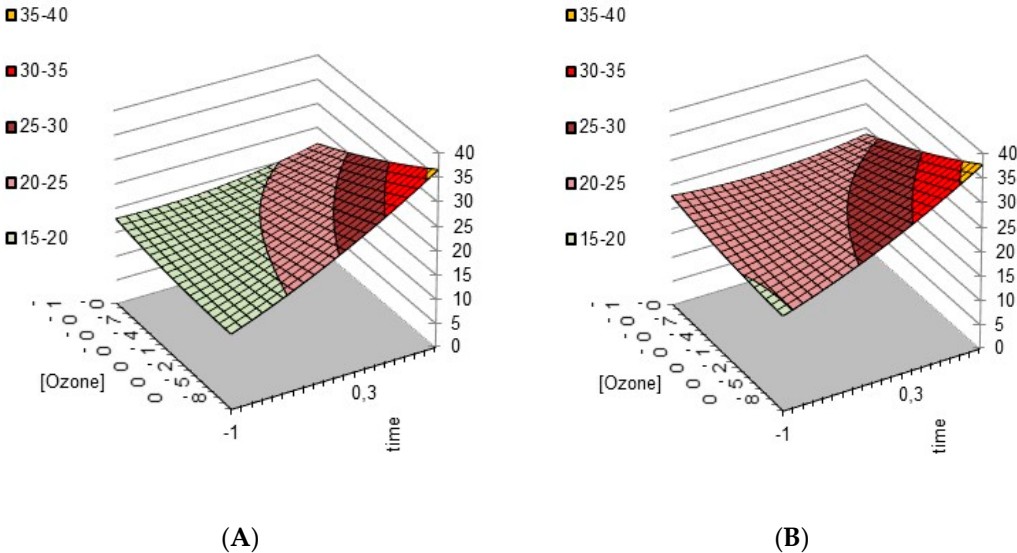

**(A)** **(B)**

**Figure 10.** Effect of pH, [O3], and reaction time on the tensile strength loss %: **(A)** pH = 3; **(B)** pH = 7.

### 3.7. Optimization of the Stripping Process

As discussed, a good stripping is generally associated with low mechanical properties. The optimal stripping process should fulfill maximum decolorization and minimum tensile strength loss. The optimum conditions have been calculated for 80% color stripping and 20% mechanical strength

loss, and this can be achieved at pH 3, using an ozone concentration of 85 g/m$^3$ and a treatment time of 10 min. Experimentally, these conditions allowed having 90% color stripping and 13% mechanical strength loss.

## 4. Discussion

In the ozonation process studied, ozone concentration and reaction time are favorable to the decolorization of the blue dyed cotton fabric. However, these conditions lead to the deterioration of the fabric sample, leading to loss in the tensile strength. The pH used plays a crucial role in the ozonation process, and as pH increases, decolorization becomes less effective and loss in mechanical strength increases. At low pH, color stripping is better and the fabric degradation is lower.

It is known that cellulosic materials get degraded upon hydrolysis of glucoside bonds in acidic conditions, since the cellulosic chain length (DP) decreases [13]. Similar results were obtained for the mechanical properties by Arooj (2014), who studied the bleaching process of cotton fabric with ozone, and showed that degradation, measured by the DP decrease of the cellulose, was greater at low pH. However, this effect occurs mainly at pH 2. In our case, the lower degradation at acidic pH would be related to the more selective ozone molecular reaction. Two possible reaction mechanisms may occur: direct reaction consisting of the attack by the molecular ozone and the indirect reaction involving the free radical mechanism. Both of these exist simultaneously during the ozonation process [14]. At low pH, ozone reacts directly in the molecular form and it is more soluble in water. However, at higher pH, the instability of ozone increases due to the reaction with hydroxyl ions OH-, thus generating free radicals. Recently, Bilińska et al. (2017) have modeled the ozonation in aqueous solution of the Reactive Black 5 dye. At low pH, there is a direct attack of the chromophoric azo groups $N = N$ by ozone [15]. The kinetics modeling shows that when pH is higher than 4.0, the indirect oxidation mechanism with free radicals starts to occur, and the discoloration kinetics are highly increased. Our results are in agreement with this model.

The optimized and highest stripping obtained in this study is, however, limited by the slight yellow coloration of the treated sample. According to Arooj (2014), aldehyde groups and ozone residues are responsible for this coloration and could be removed by further post-treatments.

## 5. Conclusions

This research points out the opportunity for color stripping cotton fabrics to be carried out with ozone dissolved in water. It quantifies the effect of ozone flow rate, time, and pH on both the decolorization and the mechanical properties.

Results show that acidic pH provided good decolorization results and less mechanical damage. Decolorization of almost 98% was obtained from the reactive-dyed textiles at pH 3, ozone concentration of 85 g/m$^3$ NTP ozone, and treatment time of 30 min. The optimization of the process has been evaluated by Box–Behnken analysis. It defines the major role of time in the degradation process and that of ozone concentration for the color stripping. An optimal set of experimental conditions is proposed.

This pilot-scale process has proven the good selectivity for ozone color stripping of a cotton fabric dyed with a blue reactive dye. However, further research should be done to avoid the slight yellow coloration of the treated sample.

Outcomes from this study suggest that the proposed ozonation treatment scheme is an efficient method for color stripping in textile processing, at pilot scale. Results will help in constructing a bulk-scale process for the textile industry.

**Author Contributions:** In this study, A.S.P., A.P., L.H., T.A. conceived and designed the experimental model, A.S.P., L.H. performed and characterized the experiments and results. All authors have discussed the results A.S.P., A.P., L.H., N.B. wrote the manuscript, A.S.P., A.P., L.H., T.A., N.B. corrected and validated the manuscript. All authors have read and agreed to the published version of the manuscript.

**Funding:** This research was funded by Erasmus Mundus program SMDTex project (Sustainable Management and Design for Textiles) which is financed by the European Commission.

**Acknowledgments:** The author would like to acknowledge the Dyeing Company for providing the dyed fabric and the UniLaSalle, Beauvais for their help and support in the ozone experiments. Also, the author would like to acknowledge the Teinturerie Lenfant, France, and the Achitex Minerva, France, for their help in dyeing the fabric and providing the reactive dyestuff, respectively.

**Conflicts of Interest:** The authors declare no conflict of interest.

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
