# Peer review of "Application of Ozone Treatment for the Decolorization of the Reactive-Dyed Fabrics in a Pilot-Scale Process—Optimization through Response Surface Methodology"

_sustainability, doi:10.3390/su12020471_

Round 1

Reviewer 1 Report

This manuscript reports that ozone treatment of reactive-dyed fabric relating to three different parameters, but there are some points to response and reconsider for revision, as followed

What is exact dye either C.I. Reactive Black 5 or reactive blue ? Because all data based on a blue dye that is not mentioned in the materials part, contrastly in materals part C.I. Reactive Black 5 was introduced. In case of ozone treatment to decompose the reactive dye molecule, there should be suggested how ozone worked to decompose the dye molecule although the analytical data could be limited. Fig. 4 & Fig. 5 : All curves except std. dyed sample & un-dyed sampe are not distinguished each other, therefore these data can be easily shown by a table rather than a figure like this. It is reuired to explain the reason why the treatment temperature was not tested as one of parameters, as the decoloration should take place by chemical reactions between oxidative ozone and reactive dye molecule thus the temperature can act more than pH conditions.  

Author Response

Dear Sir/Madam, 

Best Regards, 

Ajinkya 

Reviewer 2 Report

This paper is interesting but it has some problems in the methodology.  The comments are as follow:

1. Only an azo dye was selected in the study. Compared to other types of dye structure, azo compounds are more susceptible to oxidation. Did author try other types of dye structure? And how are the results?

2. Cotton fabric is sensitive to the low pH value. Long-time reaction with acid also could contribute to the decrease in mechanical properties. Did author identify the decrease was caused by oxidation or the acid, or both of them?

3. The pH value is very essential to the whole process, Ozone could be more efficient at low pH value in most case. Usually, temperature is another important influence factor, did the author take it into consideration?

4. The undyed cotton fabric should be considered as blank control too.

Author Response

Dear Sir/Madam, 

Best Regards, 

Ajinkya 

Round 2

Reviewer 1 Report

The revision seems to be adequate, but it is recommended to re-draw the chemical structure for C.I. Reactive Black 5.